# Graph Distributional Analytics: Enhancing GNN Explainability through Scalable Embedding and Distribution Analysis

## Abstract

Graph Neural Networks (GNNs) have achieved significant success in processing graph-structured data but often lack interpretability, limiting their practical applicability. We introduce the Graph Distributional Analytics (GDA) framework, leveraging novel combinations of scalable techniques to enhance GNN explainability. The integration of Weisfeiler-Leman (WL) graph kernels with distributional distance analysis enables GDA to efficiently quantify graph data distributions, while capturing global structural complexities without significant computational costs. GDA creates high-dimensional embeddings employing WL kernels, measures the distribution of distances from measures of categorical central tendency, and assigns distribution scores to quantify each graph's deviation from this vector We evaluate GDA on the ENZYMES, ogbg-ppa, and MalNet-Tiny datasets. Our experiments demonstrate GDA not only accurately characterizes graph distributions but also outperforms baseline methods in identifying specific structural features responsible for misclassifications. This comprehensive analysis provides deeper insights into how training data distributions affect model performance, particularly with out-of-distribution (OOD) data. By revealing the underlying structural causes of GNN predictions through a novel synergy of established techniques, GDA enhances transparency and offers a practical tool for practitioners to build more interpretable and robust graph-based models. Our framework's scalability, efficiency, and ability to integrate with various embedding methods make it a valuable addition to the suite of tools available for GNN analysis.

## 1 Introduction

Graph Neural Networks (GNNs) have emerged as a powerful framework for learning from graph-structured data, with applications ranging from social network analysis to molecular biology (Wu et al., 2020). While GNNs excel at learning complex, non-Euclidean relationships, their decision-making processes remain largely opaque (Bodria et al., 2023). This lack of interpretability poses significant challenges, especially in critical domains like bioinformatics and healthcare, where model decisions must be both accurate and explainable (Sanchez-Lengeling et al., 2020; Adoni et al., 2020). Explainability is crucial for understanding why models make specific predictions, detecting potential biases, and improving model generalization (Ju et al., 2024).

In recent years, efforts have been made to develop explainability tools tailored for GNNs. However, most existing methods focus on explaining node- or edge-level predictions, relying on gradient-based or perturbation-based approaches (Rajabi & Kafaie, 2022). While useful in some contexts, these methods struggle with graph-level predictions, particularly in datasets where structural diversity within categories complicates classification (Georgousis et al., 2021). Furthermore, many of these tools are computationally expensive and require significant customization to adapt to different datasets, limiting their scalability and practical applicability (Agarwal et al., 2022).

This paper introduces Graph Distributional Analytics (GDA), a novel framework designed to address the limitations of existing GNN explainability methods. GDA leverages Weisfeiler-Leman graph kernels to embed graph structures into high-dimensional vector spaces, allowing for the characterization of structural distributions within graph categories. By analyzing the distribution of

graph embeddings, GDA provides insights at both population and sample levels, offering a scalable and interpretable approach to understanding GNN model behavior.

Through a series of experiments on the ENZYMES, MalNet-Tiny, and ogbg-ppa datasets, we demonstrate how GDA can reveal structural anomalies and distributional shifts that contribute to model underperformance. Unlike previous explainability methods, GDA not only identifies potential sources of confusion within the dataset but also offers a preemptive tool for improving data splits and detecting out-of-distribution samples. The contributions of this paper are threefold: (1) the introduction of GDA for graph-level explainability, (2) validation of GDA's utility through case studies on real-world datasets, and (3) a discussion of how GDA can inform future research into scalable, interpretable GNN models.

## 2 RELATED WORK

Demystifying neural network decision-making is a Herculean task, particularly for non-Euclidean data, where complex interconnections stymie explainability (Adoni et al., 2020). Current explainability methods demand significant computational resources (Sanchez-Lengeling et al., 2020), and the explanations offered by these tools tend to be approximate and difficult to interpret (Li et al., 2022). If explainability tools are difficult to explain, their usefulness is questionable. Nearly all methods require extensive tailoring to specific datasets and models which reduces practicability for diverse, real-world environments (Agarwal et al., 2022). Despite increased community interest, most libraries have prioritized accuracy benchmarks over explainability (Agarwal et al., 2023).

**Gradient-based explainers**, such as Grad (Simonyan et al., 2014), Grad-CAM (Pope et al., 2019), GuidedBP (Baldassarre & Azizpour, 2019), and Integrated Gradients (Sundararajan et al., 2017), highlight specific feature importances by backpropagating gradients. These methods work well for image or node classification but struggle with graph-level predictions (Kakkad et al., 2023), which are critical for datasets like MalNet-Tiny, where node and edge features are absent.

**Perturbation-based explainers**, such as GNNExplainer (Ying et al., 2019), PGExplainer (Luo et al., 2024), SubgraphX (Yuan et al., 2021), and GraphLIME (Huang et al., 2023), evaluate how model predictions change in response to input perturbations. D4Explainer (Chen et al., 2024) offers a more efficient method by using a discrete denoising diffusion model tailored towards in-distribution explanations. While these methods offer valuable insights, they are computationally expensive and typically explore only a fraction of possible perturbations, often missing the structural nuances that impact graph-level decisions (Munikoti et al., 2023).

**Surrogate-based explainers**, like PGMExplainer (Vu & Thai, 2020), use simpler models such as Bayesian networks to approximate GNN behavior. Although they provide interpretability, they face scalability challenges when dealing with high-dimensional probability distributions, and they require substantial tuning to capture relevant patterns in large datasets (Zhu et al., 2022).

Several gaps remain in GNN explainability. First, most approaches focus on node- and edge-level tasks, neglecting graph-level explainability crucial for structural prediction problems (Pope et al., 2019). Additionally, their computational inefficiency renders them impractical for dynamic, large-scale graph environments (Kazemi et al., 2020). Finally, due to the difficulty of characterizing graph data, the literature primarily focuses on explainability for Euclidean data, which leaves non-Euclidean datasets like MalNet-Tiny underexplored (Kakkad et al., 2023).

Graph Distributional Analytics (GDA) addresses these issues by offering scalable, dataset-agnostic explanations for both population- and sample-level predictions with linear time complexity, making it suitable for real-world applications. GDA also tackles the challenge of out-of-distribution (OOD) detection, enabling better interpretation of model behavior on unseen data by characterizing graph distributions. GDA fills this gap by analyzing graph structures without relying on node or edge features, providing a more versatile approach for complex, graph-structured data.

# 3 GRAPH DISTRIBUTIONAL ANALYTICS (GDA) FOR EXPLAINABILITY

## 3.1 GRAPH EMBEDDING AND STRUCTURAL DISTRIBUTION CHARACTERIZATION

Let $G = (V, E)$ denote a graph where $V$ represents the vertices and $E$ the edges. To analyze the structural properties of graphs, we first embed each graph as a high-dimensional vector using the Weisfeiler-Leman (WL) graph kernel. For each graph, we define an embedding function $\tau : G \to \mathbb{R}^d$, where $\tau_h(G)$ represents the graph embedding after $h$ iterations of the WL kernel, and $d$ is equal to the cardinality of unique labels in $\tau_h(G)$. Formally, the embedding is given by:

$$\tau_h(G) = \sum_{v \in V} l_h(v) \quad \text{where} \quad l_h(v) = f(l_{h-1}(v), \{l_{h-1}(u) : u \in \mathcal{N}(v)\}),$$

where $l_0(v)$ is the initial label of vertex $v$, $\mathcal{N}(v)$ represents the neighborhood of $v$, and $f$ is a hash function aggregating neighborhood information (Shervashidze et al., 2011). This process captures the structural features of the graph, producing a vectorized representation that preserves graph topological properties (Morris et al., 2019). This is repeated for each $G$ in a dataset $H$ to create a set of embeddings:

$$\tau_h(H) = [\tau_h(G) \, \forall G \in H] \tag{1}$$

However, the embeddings generated are heterogeneous in both cardinality and the correspondence of mappings from labels to specific dimensions. Let $L = \{L_1, L_2, \cdots, L_n \mid n = |H|, L_j \neq L_{i<j}\}$ be the set of unique vectors generated by $\tau_h(H)$ using the WL graph kernel. To standardize the embeddings, we define the Hamel dimension, $a$, as the cardinality of the set of unique labels $L$, and construct a vector space $T$, such that $\dim(T) = T^{|L|} = \mathbb{R}^a$. A mapping $g(L) \mapsto \mathbb{R}^a$ ensures that each unique label in $L$ corresponds to a specific dimension in $\mathbb{R}^a$. For each embedding in $H$, we generate a uniform vector $\eta(G)$ as follows:

$$\forall G \in H, \eta(G) = g(\phi(G)) \in \mathbb{R}^a \tag{2}$$

The vectors within $\eta(H)$ have very high dimensionality, and many dimensions within $T$ are sparse, containing non-zero values for only a few elements in $\eta(H)$. We filter these non-informative dimensions by constructing a linear subspace $U \subset \mathbb{R}^{b \leq a}$, defined by a function $\phi(G) \mapsto \mathbb{R}^b$:

$$\forall j \in \{1, \cdots, a\}, \eta(G)_j \in \phi(G) \iff \sum_{i=1}^{n=|H|} \eta(H_i)_j \leq |H| \times \kappa \tag{3}$$

where $0 \leq \kappa \leq 1$ denotes a percentage threshold such that $0 \leq |H| \times \kappa \leq |H|$. Numerous methods exist among the feature selection literature that can be used to define $\kappa$ to optimize the removal of irrelevant dimensions from the embedding vector (Zebari et al., 2020; Gui et al., 2016). However, during the investigation of the effectiveness of GDA towards GNN explainability, we opted for a conservative approach that set $\kappa$ such that $|H| \times \kappa = \max(1, 0.002 \times |H|)$. This still significantly reduces the dimensionality of the embedding since a majority of embeddings, regardless of dataset size, are unique to one sample due to the combinatorial immensity of possible graph structures (Bondy et al., 1976).

After embedding, GDA characterizes the distribution of graph embeddings within each class. Let $\mathcal{G}_C$ represent the set of graphs belonging to class $C$. The mean embedding for class $C$ is calculated as:

$$\mu_C = \frac{1}{|\mathcal{G}_C|} \sum_{G \in \mathcal{G}_C} \phi(G),$$

which serves as the central tendency for the graphs within the class. We then calculate the cosine similarity $S_c(G)$ between each graph embedding $\phi(G)$ and the class mean $\mu_C$:

$$S_c(G) = \frac{\phi(G) \cdot \mu_C}{\|\phi(G)\|\|\mu_C\|}.$$

This similarity score quantifies how closely each graph aligns with the structural norm of its class.

To assess the degree of deviation from the class central tendency, we introduce a normalized distribution score for each graph $G$:

$$z(G) = \frac{S_c(G)}{\sqrt{\frac{1}{|\mathcal{G}_C|} \sum_{G' \in \mathcal{G}_C} S_c(G')^2}}.$$

This score reflects how much a graph diverges from its class's prototypical structure. Graphs with lower $z(G)$ values indicate significant deviations and are often associated with misclassifications or outliers. By analyzing these scores, GDA can reveal structural anomalies and patterns responsible for misclassification.

---

**Algorithm 1** Graph Embedding and Structural Distribution Characterization using Weisfeiler-Leman Kernel

---

**Input:** Dataset $H$ of graphs, iteration count $h$, threshold $\kappa$
**Output:** Set of filtered embeddings $\eta(H)$ and distribution scores $z(G)$ for each $G \in H$
**for** each graph $G = (V, E) \in H$ **do**
    Initialize node labels $l_0(v)$ for all $v \in V$
    **for** $i = 1$ to $h$ **do**
        **for** each vertex $v \in V$ **do**
            Update label: $l_i(v) = f(l_{i-1}(v), \{l_{i-1}(u) \mid u \in \mathcal{N}(v)\})$
        **end for**
    **end for**
    Compute graph embedding: $\tau_h(G) = \sum_{v \in V} l_h(v)$
**end for**
Let $L = \{L_1, L_2, \cdots, L_n \mid n = |H|, L_j \neq L_{i<j}\}$ represent the set of unique labels across all graphs
Define vector space $T \in \mathbb{R}^a$ where $a = |L|$
**for** each graph $G \in H$ **do**
    Map embedding: $\eta(G) = g(\tau_h(G)) \in \mathbb{R}^a$
**end for**
**for** each dimension $j \in \{1, \cdots, a\}$ **do**
    **if** $\sum_{i=1}^{|H|} \eta(H_i)_j \leq |H| \times \kappa$ **then**
        Remove dimension $j$ from all embeddings $\eta(G) \in H$
    **end if**
**end for**
Compute the filtered embeddings $\eta(H) \in \mathbb{R}^b$ where $b \leq a$
**for** each class $C$ in dataset $H$ **do**
    Compute class mean embedding: $\mu_C = \frac{1}{|\mathcal{G}_C|} \sum_{G \in \mathcal{G}_C} \eta(G)$
    **for** each graph $G \in \mathcal{G}_C$ **do**
        Compute cosine similarity: $S_c(G) = \frac{\eta(G) \cdot \mu_C}{\|\eta(G)\|\|\mu_C\|}$
        Compute normalized distribution score: $z(G) = \frac{S_c(G)}{\sqrt{\frac{1}{|\mathcal{G}_C|} \sum_{G' \in \mathcal{G}_C} S_c(G')^2}}$
    **end for**
**end for**
**Output:** Filtered embeddings $\eta(H)$ and distribution scores $z(G)$ for each $G \in H$

---

## 3.2 Outlier Detection and Distribution Analysis

In the context of Graph Distributional Analytics (GDA), we define outliers as graphs whose structural properties significantly deviate from the central tendency of their class. Using the embeddings

generated by the Weisfeiler-Leman (WL) kernel, we quantify these deviations based on cosine similarity between the graph embeddings and the mean embedding of the class. Additionally, we assess abnormal distributions within classes using kurtosis, a measure of how heavily tailed a distribution is compared to a normal distribution.

### 3.2.1 OUTLIER DETECTION

We define outliers as graphs whose similarity score falls significantly below the average similarity for the class. Specifically, a graph $G \in \mathcal{G}_C$ is considered an outlier if:

$$S_c(G) < \mu_{S_c} - \alpha \sigma_{S_c},$$

where $\mu_{S_c}$ and $\sigma_{S_c}$ are the mean and standard deviation of cosine similarity scores across all graphs in class $C$, and $\alpha$ is a user-defined threshold (typically set to 2 or 3) that determines how many standard deviations away from the mean qualifies as an outlier. This method ensures that graphs with significantly lower similarity to the class mean are identified as outliers, which can help explain misclassifications or structural anomalies in the dataset.

### 3.2.2 ABNORMAL DISTRIBUTION DETECTION VIA KURTOSIS

To detect abnormal distributions within a class, we use kurtosis, $\xi(G_c)$ which measures the "tailedness" of the distribution of cosine similarity scores. High kurtosis indicates that the distribution has heavy tails, meaning there are more extreme outliers than expected under a normal distribution. Low kurtosis suggests that the distribution has lighter tails, indicating fewer extreme values.

Kurtosis for a set of cosine similarity scores $\{S_c(G) \mid G \in \mathcal{G}_C\}$ is defined as:

$$\xi(G_c) = \frac{1}{n} \sum_{i=1}^{n} \left( \frac{S_c(G_i) - \mu_{S_c}}{\sigma_{S_c}} \right)^4 - 3,$$

where $\mu_{S_c}$ and $\sigma_{S_c}$ are the mean and standard deviation of the similarity scores, and $n = |G_C|$ is the number of graphs in class $C$. A kurtosis value greater than 3 indicates a distribution with heavy tails, suggesting that the class contains a significant number of structural outliers or subgroups with distinct structural features. Conversely, a kurtosis value less than 3 indicates a distribution with light tails, where most graphs are similar to the class mean.

By analyzing kurtosis, GDA can identify classes with abnormally high or low variability in graph structures, which can provide insights into model performance. For instance, classes with high kurtosis may indicate the presence of substructures that cause misclassifications or other performance issues, while low kurtosis may suggest that the class is highly homogeneous and thus easier for the model to learn.

## 3.3 POST-HOC STRUCTURAL ATTRIBUTION FOR MISCLASSIFIED GRAPHS

For misclassified graphs, GDA provides a post-hoc explanation by tracing structural deviations. By rerunning the WL kernel with degree sequence tracking, we can identify specific substructures responsible for classification errors. The process involves examining how node labels evolve through each iteration, enabling us to pinpoint graph substructures that deviate from the class norm. This structural attribution mechanism allows for an interpretable analysis of misclassifications, offering insights into how structural features influence GNN predictions.

## 3.4 SCALABILITY AND INTEGRATION WITH EXISTING EXPLAINABILITY METHODS

GDA's key strength lies in its scalability. The embedding and distribution analysis have a computational complexity of $O(n \cdot m)$, where $n$ is the number of graphs and $m$ is the average number of nodes per graph. This ensures that GDA can handle large-scale graph datasets efficiently.

Moreover, GDA complements existing gradient-based and perturbation-based explainability techniques. While Grad-CAM and GNNExplainer focus on the node or edge-level importance, GDA

| Dataset | Avg. # Nodes | Avg. # Edges | Node Features | Edge Features | # Categories |
|---------|-------------|-------------|---------------|---------------|--------------|
| MalNet-Tiny | 1,410.3 | 2,647.3 | N/A | N/A | 5 |
| ogbg-ppa | 243.4 | 2,266.1 | N/A | 6 features | 37 |
| ENZYMES | 32.6 | 62.1 | Atom type | N/A | 6 |

Table 1: Dataset Statistics
Statistics highlighting graph order, graph size, number of classification categories, and the presence or absence of node features.

offers a higher-level, graph-wide perspective. This allows GDA to serve as a versatile framework, providing insights at both population and sample levels, including outlier detection and out-of-distribution (OOD) analysis.

## 4 EXPERIMENTS

The experiments conducted in this study validate the utility of Graph Distributional Analytics (GDA) for explaining GNN performance. GDA provides insights into model behavior and data distributions, but it does not prescribe specific actions to improve model performance. The results presented in this section demonstrate how GDA can inform further analysis and lead to improvements, though the implied action remains open and is subject to further research.

We present specific case studies that highlight the power of GDA at both population and sample levels. The body of this paper focuses upon how GDA is employed for GNN explainability. Detailed quantitative accountings of experiments are included in the appendices.

### 4.1 EXPERIMENTAL SETUP

All experiments were conducted by both models on all three datasets for ten runs with seeds (numpy, torch, random) set to $\{1, 2, \cdots, 10\}$ corresponding to the run of the experiment. Initial experiments, also referred to as baseline experiments, used prescribed training, test, and validation splits as defined by the dataset.

#### 4.1.1 DATASETS:

Three datasets were examined for this study:

- **MalNet-Tiny:** MalNet-Tiny (Freitas & Dong, 2021) is a curated subset of 5000 of the MalNet dataset graphs where each graph is selected to have fewer than 5000 nodes. Both datasets seek to classify function calls as indicative of malware or benign. It is designed for rapid prototyping of methods and seeks to strike a balance between the problem difficulty of the larger dataset and the computational demands of processing large data.

- **ogbg-ppa:** ogbg-ppa (Szklarczyk et al., 2019) is a graph-structural prediction dataset focused on identifying protein interactions between different species and taxa of organisms from the Stanford Open Graph Benchmark (Hu et al., 2020).

- **ENZYMES:** ENZYMES (Schomburg et al., 2004) is a part of the larger TUDataset benchmark (Morris et al., 2020). It seeks to identify protein tertiary structures categorized into six enzyme classes, with graphs representing the structural relationships between amino acids.

#### 4.1.2 MODELS

To fully place the emphasis of the study upon the application of GDA for GNN explainability, we examined all datasets using the same model architectures to ensure relevance and comparability.

- **GraphSAGE:** A scalable, inductive graph learning model that generates embeddings by sampling and aggregating features from a node's local neighborhood. This model is particularly useful for learning representations in large-scale graphs (Hamilton et al., 2017).

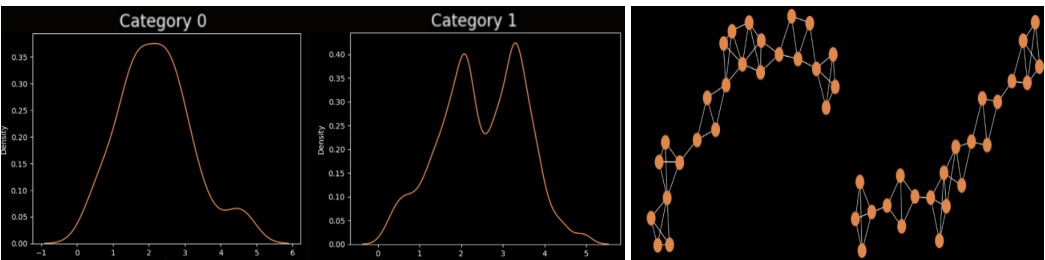

Figure 1: Abnormal Distribution for Transferases

LEFT: Category 0 demonstrates a 'nice' distribution with low kurtosis, but Category 1 has a strange double peak indicative of a bimodal distribution for the transferases category of the Enzymes dataset. This double-peak demonstrated high kurtosis which indicated further investigation was warranted.

RIGHT: Here we observe two distinctly different graph structures both categorized as transferases which originates from the enzymes functional (not structural) definition. Separating this class during training may help prevent model confusion when trying to structurally classify these samples.

*Architecture:* The architecture consists of three SAGEConv layers with hidden channels. An AttentionalAggregation layer is used with a gate neural network that includes a linear layer and a sigmoid activation. This is followed by two fully connected layers where the hidden channels are reduced by half in the first layer and then mapped to the number of output classes in the second layer.

- **Graph Isomorphism Network (GIN):** A powerful GNN model designed to distinguish graph structures more effectively by leveraging an aggregation function that closely resembles the Weisfeiler-Lehman graph isomorphism test (Xu et al., 2019).

  *Architecture:* The architecture consists of one initial GINConv layer, which maps the input features to the hidden channels, followed by three additional GINConv layers. Each GINConv block includes two linear layers, ReLU activations, and batch normalization. After the GINConv layers, a global mean pooling operation aggregates the node embeddings into a graph-level representation. The final classification is performed using two fully connected layers, with dropout applied between them.

## 4.2 RESULTS

In this section, we present a comprehensive analysis of how Graph Distributional Analytics (GDA) was applied across multiple datasets to examine structural anomalies and their impact on model performance. The experiments focus on understanding population-level distribution shifts and sample-level misclassifications in the ENZYMES, MalNet-Tiny, and ogbg-ppa datasets. All experiments were conducted using the GraphSAGE and GIN architectures to maintain consistency and comparability across datasets.

### 4.2.1 POPULATION LEVEL ANALYSIS

The population-level analysis begins with the ENZYMES dataset, where GDA was applied to detect abnormal distribution patterns within the graph embeddings. A key observation in this dataset was the bimodal distribution found in Category 1 that presented a high kurtosis value, corresponding to the transferase enzymes. Transferases are a functional definition, not a structural definition, so while assigned to the same category, they exhibit significant structural heterogeneity. The existence of two distinct clusters within this category suggested that the enzyme structures were quite different, despite their shared functional role. This bimodal distribution, which was confirmed by the structural representations are shown in Figure **??**. This finding is consistent with biochemical literature that highlights the variability within enzyme families (Giegé et al., 2012; Breton et al., 2006). The structural diversity in transferases led to underperformance in this category, with the GNN models struggled to capture the dual signals introduced by these two clusters.

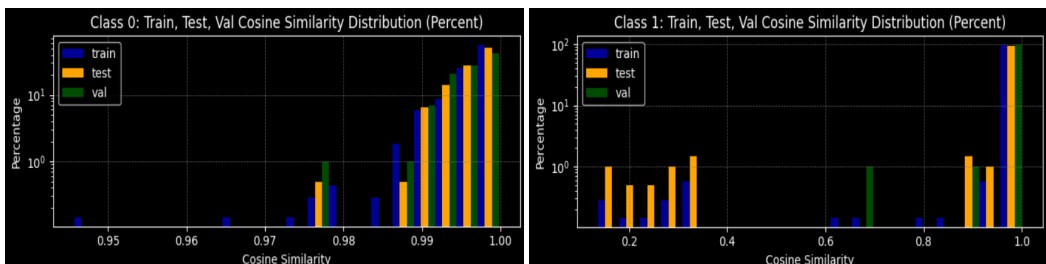

Figure 2: Distribution Shifts in Malnet-Tiny

In Class 0 (left), we see a relatively normal distribution of similarities for a classification category with nearly all samples lying close to the central tendency vector. However, in Class 1 (right), we see an atypical distribution that has a significant portion of samples being nearly orthogonal to the central tendency.

To address this issue, we computed Pearson correlation coefficients on the embedding vectors, which allowed us to separate the two clusters and treat them as distinct categories during model training. After the training phase, we recombined the categories, leading to a 2.3% performance improvement for the transferase category, which also translated to a 0.4% increase in overall dataset performance across both models repeated with ten separate seeds. These results highlight the importance of identifying and handling structural heterogeneity within a class, which can significantly impact model accuracy. This observation aligns with prior work showing that structural complexity and variability within functional groups can lead to model confusion (Chen & Wu, 2022; Ji et al., 2021).

Moving to the MalNet-Tiny dataset, we found that three out of five categories exhibited relatively normal distributions with cosine similarities clustered near 1 and low kurtosis. However, two categories showed significant divergence, with some graph embeddings nearly orthogonal to their class mean vectors, as seen in Figure 2. These outliers indicated the presence of structurally distinct graphs that deviated from the majority of samples in their respective categories. Our hypothesis was that these outliers were injecting noise into the model, leading to poorer performance in those categories. To test this, we removed the outliers and retrained the models. While the improvements were modest, the reduction in false positives was notable. We hypothesize that the model was already learning to ignore these outliers during training in the baseline runs, as their removal only reflected an equivalent reduction in false negative samples. This observation reinforces the idea that models can effectively disregard noisy samples under certain conditions, but identifying and removing them still contributes to cleaner classification metrics.

A related experiment involved analyzing distribution shifts between the training, validation, and test sets in the MalNet-Tiny dataset. GDA revealed significant shifts in the distributions, with the training set embeddings clustering more tightly around the mean compared to the test and validation sets. These shifts, shown in Figure 3, likely caused overfitting, as the model was able to perform well on the more homogeneous training set but struggled to generalize to the more diverse test and validation samples. To address this, we restructured the dataset splits to ensure more consistent distributions across the sets. After retraining the models, we observed an average 4.3% improvement in performance over the baseline across 10 runs, using both GraphSAGE and GIN architectures. These results emphasize the critical importance of ensuring distributional consistency between dataset splits to prevent overfitting and improve generalization, as noted in previous studies on distributional shifts in machine learning (Ding et al., 2021; Fan et al., 2024).

### 4.2.2 SAMPLE LEVEL ANALYSIS

Finally, we applied GDA to analyze individual misclassifications at the sample level, particularly in the ogbg-ppa dataset. In this case, GDA identified a structural motif (Figure 4) that was present in 12% of misclassified samples from Category 5 but was prevalent in 68% of Category 27 samples. This structural overlap likely led to misclassifications, as graphs containing this motif were frequently classified as belonging to Category 27, even when they were from Category 5. Misclassification due to structural imbalances is a well-known issue in graph-based learning systems, and similar findings have been reported in the literature (Ding et al., 2021). By isolating this motif and

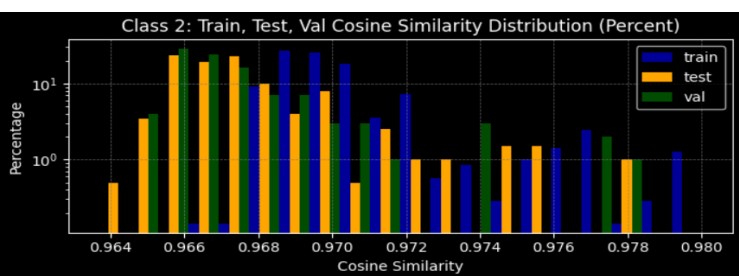

Figure 3: Distribution Shifts
In classification category 2 of the MalNet-Tiny dataset, we see a significant shift between the training distribution, which is closer to the central tendency, and the testing and validation shifts which tend to be less similar. This may result in overfitting by the model due to the disparity between training and testing/validation distributions.

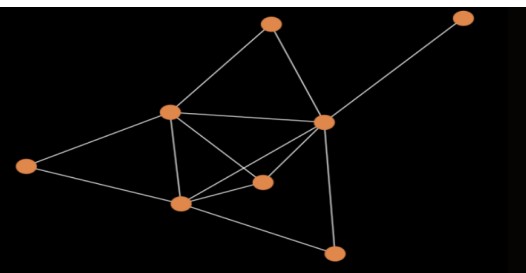

Figure 4: Embedding Analysis
This structure was present in a majority of misclassified examples in one class. We discovered it is a highly prevalent structure in another class which may have led to models becoming stuck in local minima when trying to differentiate these samples.

adjusting the training process to account for this structural overlap, we observed a significant reduction in misclassification rates between the two categories. This case study demonstrates the utility of GDA in identifying structural imbalances and mitigating their effects on model performance.

While benchmark datasets like those used in this study are useful for prototyping, their simplicity precludes in-depth study of individual samples. The sample-level analysis provided by GDA is most powerful when employed by those with domain expertise. The lack of detailed information in protein, enzyme, and malware identification datasets, which would be required to tie structures to real-world examples, hindered the application of this method in this study..

## 5  CONCLUSION

Graph Distributional Analytics (GDA) offers a novel and scalable framework for understanding the structural complexities that influence model performance in Graph Neural Networks (GNNs). By leveraging Weisfeiler-Leman graph kernels to embed graphs and analyze their distributions, GDA provides insights at both population and sample levels. The experiments conducted on the ENZYMES, MalNet-Tiny, and ogbg-ppa datasets demonstrate that GDA can effectively identify structural anomalies, distributional shifts, and misclassified samples that traditional methods may overlook. While GDA does not prescribe specific solutions, it offers a powerful tool for preemptive data analysis, improving dataset splits, and detecting out-of-distribution samples. Future work will focus on integrating GDA with more systematic model refinement strategies and further exploring its applications in real-world GNN tasks.

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

## A  APPENDIX

### A.1  EXPERIMENTAL RESULTS

#### A.1.1  ENZYMES-BEFORE AND AFTER SPLITTING CLUSTERS

| Run | 1 | 2 | 3 | 4 | 5 | 6 | 7 | 8 | 9 | 10 | Avg |
|---|---|---|---|---|---|---|---|---|---|---|---|
| GraphSAGE | | | | | | | | | | | |
| Before | 83.3 | 80.6 | 83.0 | 81.8 | 81.9 | 80.1 | 83.5 | 82.3 | 82.8 | 81.7 | 82.1 |
| After | 82.8 | 82.8 | 83.7 | 82.5 | 83.5 | 81.3 | 83.9 | 83.4 | 82.9 | 80.2 | 82.7 |
| GIN | | | | | | | | | | | |
| Before | 83.3 | 81.4 | 83.8 | 80.4 | 84.9 | 83.2 | 84.9 | 82.7 | 84.2 | 82.5 | 83.1 |
| After | 82.0 | 81.8 | 83.1 | 84.4 | 82.1 | 84.7 | 84.5 | 81.7 | 84.6 | 84.4 | 83.3 |

#### A.1.2  MALNET-TINY DATASET-BEFORE AND AFTER REMOVING OUTLIERS

| Run | 1 | 2 | 3 | 4 | 5 | 6 | 7 | 8 | 9 | 10 | Avg |
|---|---|---|---|---|---|---|---|---|---|---|---|
| GraphSAGE | | | | | | | | | | | |
| Before | 84.0 | 82.7 | 81.4 | 85.2 | 81.9 | 79.1 | 79.0 | 79.2 | 82.4 | 82.8 | 81.8 |
| After | 81.1 | 81.0 | 81.1 | 82.0 | 82.7 | 82.8 | 83.1 | 81.6 | 83.4 | 81.1 | 82.0 |
| GIN | | | | | | | | | | | |
| Before | 83.5 | 87.1 | 83.2 | 81.0 | 86.1 | 87.5 | 84.6 | 86.3 | 86.7 | 86.5 | 85.2 |
| After | 86.0 | 84.8 | 83.8 | 85.0 | 86.3 | 86.9 | 85.7 | 85.4 | 85.9 | 84.3 | 85.4 |

### A.1.3   MALNET-TINY: BEFORE AND AFTER CORRECTING DISTRIBUTION SHIFTS

| Run | 1 | 2 | 3 | 4 | 5 | 6 | 7 | 8 | 9 | 10 | Avg |
|---|---|---|---|---|---|---|---|---|---|---|---|
| GraphSAGE | | | | | | | | | | | |
| Before | 84.0 | 82.7 | 81.4 | 85.2 | 81.9 | 79.1 | 79.0 | 79.2 | 82.4 | 82.8 | 81.8 |
| After | 84.1 | 87.1 | 86.4 | 85.3 | 87.6 | 87.6 | 83.9 | 84.9 | 87.3 | 85.1 | 85.9 |
| GIN | | | | | | | | | | | |
| Before | 83.5 | 87.1 | 83.2 | 81.0 | 86.1 | 87.5 | 84.6 | 86.3 | 86.7 | 86.5 | 85.2 |
| After | 88.4 | 89.0 | 89.5 | 88.8 | 90.0 | 89.6 | 89.8 | 89.8 | 90.1 | 89.1 | 89.4 |

### A.2   EXAMPLE VISUALIZATION OF FUNCTION CALL GRAPH FROM MALNET-TINY

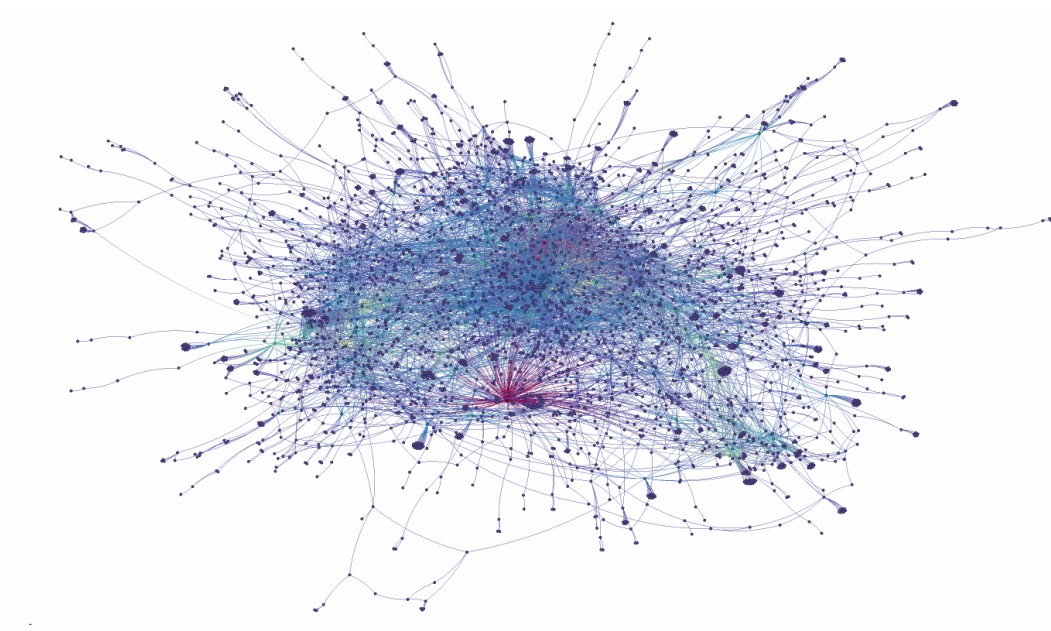

### A.3   VISUALIZATION OF EMBEDDING VECTOR IN MALNET-TINY

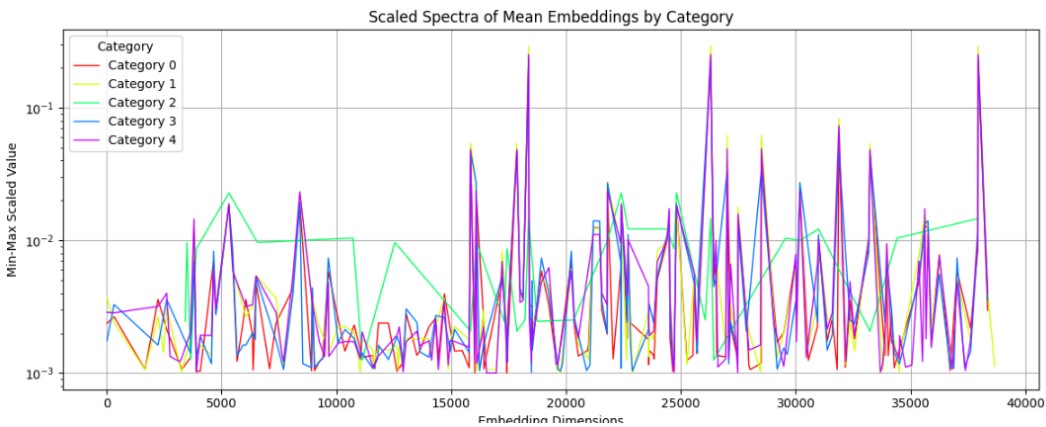

### A.4   TIME COMPLEXITY ANALYSIS OF GDA

A detailed analysis of the time complexity for the Graph Distributional Analytics (GDA) process is essential for understanding its computational efficiency.

### A.4.1 WEISFEILER-LEMAN EMBEDDING PROCESS

For each graph $G_i = (V_i, E_i)$ in the dataset $H$, the Weisfeiler-Leman (WL) embedding process is performed. The embedding involves iterating over the graph's nodes, refining their labels through the WL kernel, and aggregating these labels to form the final graph embedding. The time complexity for this embedding process is given by:

$$T_{\text{WL}}(G_i) = 5h \cdot |V_i| + 1 \tag{4}$$

where: - $h$ is the number of iterations of the WL kernel, - $|V_i|$ is the number of nodes in graph $G_i$.

This accounts for the operations required to hash and sort node labels during each iteration.

### A.4.2 MEAN VECTOR CALCULATION

The calculation of mean vectors for each classification category involves summing the embeddings of all graphs within the same category. Let: - $m$ represent the number of classification categories, - $|H|$ represent the total number of graphs in the dataset $H$.

The time complexity for computing the mean vectors is:

$$T_{\text{mean}} = m + |H| \tag{5}$$

This reflects the time required to iterate over all graphs and categories to compute their respective mean embeddings.

### A.4.3 COSINE SIMILARITY CALCULATION

Determining the cosine similarity for each graph with respect to its classification category mean vector is a key step in GDA. The time complexity for this operation across all graphs is:

$$T_{\text{cosine}} = m + 2|H| \tag{6}$$

This accounts for the multiplication and summation operations necessary to compute the cosine similarity between the graph embeddings and the category mean vectors.

### A.4.4 DISTRIBUTION SCORE CALCULATION

Calculating the distribution score for each graph involves standardizing its cosine similarity relative to the category's standard deviation. The time complexity for this step is:

$$T_{\text{score}} = m + 3|H| \tag{7}$$

This reflects the additional operations required for calculating and applying the standard deviation to each graph's cosine similarity.

### A.4.5 OVERALL TIME COMPLEXITY

The total time complexity for the entire GDA process is the sum of the complexities of each step:

$$T_{\text{GDA}} = (5h + 1) \sum_{i=1}^{|H|} |V_i| + 6|H| + 3m \tag{8}$$

Here, $\sum_{i=1}^{|H|} |V_i| = |V_H|$ represents the total number of nodes across all graphs in the dataset $H$. In the worst-case scenario, the time complexity is dominated by the term $|V_H|$, leading to an overall complexity of:

$$T_{\text{GDA}} \in O(|V_H|) \tag{9}$$

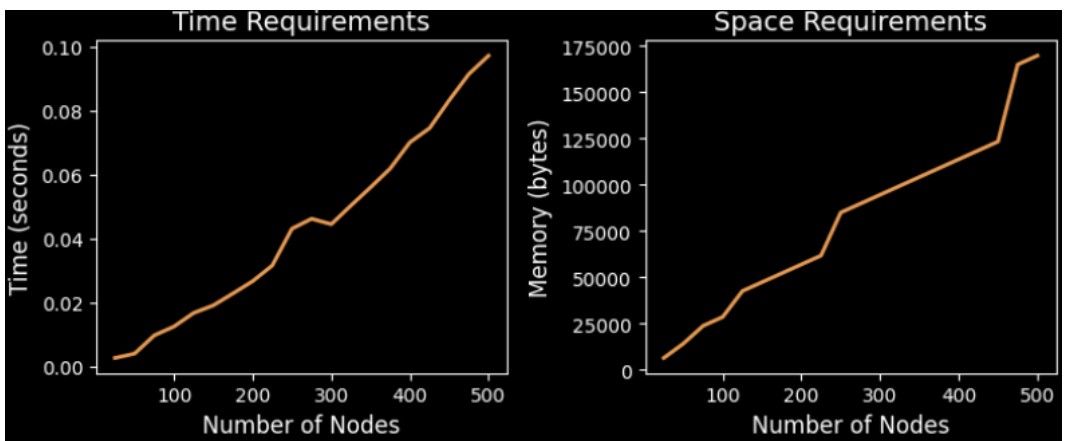

Figure 5: Time and Space Empirical Analysis of GDA

### A.4.6 STORAGE COMPLEXITY

The storage requirements for the graph embeddings are also an important consideration. Each embedding vector is normalized to a standard length based on the number of unique labels in the dataset. If every node in the dataset has a unique label, the worst-case storage requirement is:

$$S_{\text{GDA}} = |H| \times |V_H| \tag{10}$$

However, in practice, the storage requirement is often significantly smaller. For instance, in the ogbg-ppa dataset, which contains $|H| = 158,100$ graphs with an average of $|V_i| = 243$ nodes each, the length of the embedding vectors was only 2,021. This indicates that the practical storage needs are much less than the theoretical worst case.

### A.4.7 EMPIRICAL VALIDATION

The time and storage complexities were empirically validated using the datasets discussed in this paper. As shown in Figure 5, the observed time complexity closely matches the predicted linear growth with respect to the sum of the graph orders in the dataset, confirming the efficiency of the GDA approach.

### A.5 HARDWARE AND SOFTWARE SPECIFICATIONS

All experiments were conducted using Python with several key libraries integral to the development and evaluation of our models:

- **PyTorch:** The primary deep learning library used for implementing neural network models.
- **PyTorch Geometric:** A specialized extension of PyTorch designed for working with graph-structured data, crucial for implementing the Graph Neural Networks (GNNs) evaluated in this study.
- **NumPy:** Used for efficient numerical computations, particularly for handling matrix operations and dataset preprocessing.
- **Matplotlib:** Utilized for generating the figures and plots presented in the main text, providing clear visualizations of the experimental results.

All computations were performed on a cloud-based server with the following hardware specifications:

- **2 NVIDIA T4 GPUs:** These GPUs are optimized for deep learning tasks, each equipped with 16 GB of GDDR6 memory, providing the necessary computational power for training complex GNN models on large datasets.
- **30 GB RAM:** This amount of system memory facilitated the handling of large datasets and the execution of memory-intensive operations, particularly during the preprocessing and training phases.
- **8 vCPUs:** Virtual CPUs were used to manage concurrent processing tasks efficiently, ensuring smooth operation and timely completion of the experiments.
- **Ubuntu 20.04 LTS:** The operating system provided a stable and secure environment for running the software libraries and managing computational tasks.

These hardware and software configurations ensured that the experiments were executed efficiently and that the results were both reliable and reproducible. The combination of PyTorch, PyTorch Geometric, and advanced hardware like the NVIDIA T4 GPUs enabled the successful implementation and evaluation of the GDA framework across the various datasets and models used in this study.

