# OpenReview forum: "Graph Distributional Analytics: Enhancing GNN Explainability through Scalable Embedding and Distribution Analysis"
_ICLR.cc/2025/Conference — ICLR 2025 Conference Withdrawn Submission_

### Official Review · Reviewer_irWd · 2024-11-02

**Soundness:** 1
**Presentation:** 2
**Contribution:** 2
**Rating:** 3
**Confidence:** 5

**Summary:**

The paper introduces the Graph Distributional Analytics  framework to enhance the interpretability of Graph Neural Networks

**Strengths:**

The paper presents a novel approach to addressing the challenge of OOD detection by characterizing graph distributions.

**Weaknesses:**

1. The paper lacks detailed explanations, and the notation used is disorganized and confusing.
2. The evaluation is insufficient, as the paper only provides a limited number of examples to illustrate the work.
3. The presented method is not well-supported, making it difficult to grasp the intuition behind the methodology; refer to the detailed questions for clarification.
4. The computational complexity analysis on line 266 is incorrect. The complexity of the WL kernel should be linear in $n^2$, as shown in [1].



[1]Shervashidze, Nino et al. “Weisfeiler-Lehman Graph Kernels.” *J. Mach. Learn. Res.* 12 (2011): 2539-2561.

**Questions:**

1. Why choose the WL graph kernel for obtaining graph embeddings instead of a method like GNN? Kernels are generally used to measure the similarity between pairs of data points and are not typically used for generating representations. Given that the method is feasible, how does the number of iterations in the kernel affect the outcome, since different iteration counts can lead to significantly different representations for each data point?
2. As mentioned in line140, $\phi(G)$ is used for dimensionality reduction. How is $\phi(G)$ derived? Is it pre-defined, or is it adaptively chosen based on each dataset?
3. In section 3.2.2, it is mentioned that a large kurtosis value indicates that the data is very close to one class while far from another. Why does this suggest the presence of substructures that contribute to misclassifications? How can these substructures be identified?

---

### Official Review · Reviewer_bPXg · 2024-11-02

**Soundness:** 1
**Presentation:** 2
**Contribution:** 1
**Rating:** 3
**Confidence:** 3

**Summary:**

The article introduces a method for gaining insights into graph-level classification problems. This method relies on graph scores that indicate the degree of similarity between a given graph and the average representation of its corresponding class. The class mean and similarity are measured using the sample average and cosine similarity of graph embeddings.

**Strengths:**

- The tackled problem is relevant.

**Weaknesses:**

- The proposed method is not convincing. It looks like a lot of arbitrary choices are put together to define a score. The range and scale of the resulting scores is not indicative of any potential issues. Conclusions are drawn from arbitrary thresholds or qualitative considerations.
- The paper suffers from poor presentation, with unclear notation and inappropriate terminology at times.
- The proposed method is limited to graph-level classification tasks with unattributed graphs.

**Questions:**

- While handling graphs without node or edge labels can be useful in some scenarios, it’s unclear why the authors restricted the method to this limited class of graphs. Is there a specific difficulty in extending it to attributed graphs?
- How crucial is the specific embedding and scoring mechanism used here? It seems that any sufficiently rich graph embedding might work equally well, if not better. Given this, it’s unclear to what extent the proposed method offers a principled or original solution. Could you elaborate?
- What is the rationale behind the choice of $\kappa$ in line 150?
- Line 237. Why would a normal distribution be expected here?
- Line 246. Why is the threshold of 3 an appropriate choice? Is there a specific justification for this value?
- Could you please clarify what "tracing structural deviations" and "degree sequence tracking" mean in line 258?
- The stated computational complexity appears to be incorrect. The complexity grows with the order h of the WL test as high-order relations are considered. Am I missing something?


A few examples of lack of mathematical rigor and notational issues that should be addressed.
- Line 114. Function $\tau$ is introduced with an image space of dimension $d$ that depends on the input (as state in line 129), making this definition invalid. However, the hash function defined in line 118 appears to produce fixed-size representations of $G$. Could you clarify this inconsistency?
- Line 129. The term "heterogeneity" needs clarification. Additionally, could you specify the intended meaning of "cardinality of a vector"?
- Line 130. It’s evident that $L_1,\dots,L_n$ are not necessarily all unique. It seems the author intended to define $L=\{L_1,\dots,L_n\}$ by removing repeated elements, which is somewhat confusing. If $L$ is defined as a set, it does not contain duplicates by definition.
- Line 132. it is said that $a=|L|$, so it’s unclear why the vector space $T$  is immediately defined as $\mathbb R$. Could you clarify the necessity of this extra notation?
- Line 133. The definition of $g$ is inappropriate. Do you mean $g: L\to \mathbb R^a$?
- Eq 3. The notation $\eta(G)_j$ is not explained. Additionally, the use of the symbol $\in$ with a vector instead of a set is unusual; could you clarify its intended meaning in this context?

---

### Official Review · Reviewer_rZyg · 2024-11-03

**Soundness:** 2
**Presentation:** 2
**Contribution:** 2
**Rating:** 3
**Confidence:** 3

**Summary:**

This paper proposes a Graph Distributional Analytics (GDA) framework, which combines Weisfeiler-Leman graph kernels with distributional distance analysis to efficiently capture global graph structures. Tested on multiple datasets, GDA identifies structural features linked to misclassifications, offering insights into how data distributions impact GNN performance, particularly with out-of-distribution data. Its scalability and compatibility with various embeddings make GDA a valuable tool for building more interpretable, robust GNN models.

**Strengths:**

1. The proposed Graph Distributional Analytics (GDA) framework addresses key limitations in the explainability of Graph Neural Networks (GNNs), presenting a promising solution to this important problem.
2. The experimental results demonstrate that GDA provides graph-level explainability with practical utility for real-world tasks.
3. The writing in this paper is clear, well-structured, and easy to read and follow.

**Weaknesses:**

1. The theoretical contribution of this paper is weak; the proposed GDA method lacks theoretical motivation and analysis, relying solely on heuristic insights.
2. The datasets used are limited, and Table 1 does not indicate the number of graphs in each dataset. Since GDA emphasizes scalability, it would be beneficial to consider larger graphs.
3. The experiments do not include other GNN explainability methods as baselines for comparison with the proposed GDA. Including relevant baselines could enhance the quality of the results.
4. The figures in this paper are unclear, with all of them presented on a black background, making them difficult to read and understand.
5. There are minor errors in the paper that should be addressed, such as the missing figure reference in line 375.
6. The paper lacks code for reproducibility, and further clarification is needed on the experimental settings, including details like the number of model layers, learning rate, and other hyperparameter configurations.

**Questions:**

1. Please begin by clarifying the weaknesses section.
2. How is the set of unique labels L in Algorithm 1 derived? The paper indicates that they are generated using the WL graph kernel, but could the authors provide further explanation and detail on this process?

---

### Official Review · Reviewer_Mkkk · 2024-11-03

**Soundness:** 3
**Presentation:** 3
**Contribution:** 2
**Rating:** 5
**Confidence:** 4

**Summary:**

This paper proposes an extensible Graph Distributional Analytics (GDA) framework, which uses Weisfeiler-Leman ( WL ) graph kernel and distribution distance analysis to integrate quantitative graph data distribution. The importance of GDA for graph distribution analysis and its improvement on graph classification tasks are verified in three datasets and two benchmark models.

**Strengths:**

1、The GDA method of graph distribution is proposed and the overall process is clear. The overall and node level prediction is used to solve various types of graph data.
2、At the same time, the scheme has linear time complexity and has the ability to process large-scale graph datasets.

**Weaknesses:**

1. The proposed GDA algorithm is used at the data level, mainly using different quantitative analysis of the WL algorithm on the graph structure. The design source of the entire GDA scheme does not have a sufficient mathematical proof or theoretical analysis.

2. The overall experiment has a general effect on the improvement of GNN graph classification, and it is necessary to analyze and process each graph data separately with the prior knowledge of graph data.

3. The label of the 375 line icon does not correspond.

**Questions:**

1. The paper mentions the challenge of solving out-of-distribution ( OOD ) detection. By characterizing the distribution of the graph, the model behavior on the unseen data can be better explained. How is this reflected in the experiment?

2. The explanatory perspective on GNN in the paper is relatively common and needs to be further proved. The article starts from the GNN data instead of GNN structure, and analyzes the distribution of graph data. In essence, is this similar to the enhancement operation on the data?

3. As the input of the GDA, what's the meaning of the initial label of the graph node set? Could you give more explanation? And how to deal with graph node sets which don not have true label?

---

### Official Review · Reviewer_eEhC · 2024-11-04

**Soundness:** 2
**Presentation:** 2
**Contribution:** 1
**Rating:** 3
**Confidence:** 4

**Summary:**

The paper presents a method termed Graph Distributional Analytics (GDA), which first leverages WL kernels for embedding, followed by distributional distance analysis, to provide insights into structural characteristics that might impact GNN predictions. In the experiments, GDA is demonstrated to uncover structural nuances whcih may influence model performances.

**Strengths:**

The WL embedding is intuitive and scalable, and is seen to effectively uncover structural nuances.  The framework's linear time complexity makes it suitable for large-scale datasets, addressing a common limitation in existing explainability methods for GNNs.

**Weaknesses:**

The proposed method for generating Weisfeiler-Leman (WL) hash-based embeddings for graphs is intuitive and has potential for identifying structural distributional characteristics. This approach could be useful for detecting anomalies and creating better data splits to minimize the degree of structural out-of-distribution (OOD) scenarios in the test set. However, I question its utility as a true tool for GNN explainability; at best, it can indicate cases where WL-compliant GNNs are likely to struggle.

If the framework is indeed intended as an explainability tool, a comparison with existing explainability methods is essential, along with a clear discussion on the improvements this proposal offers over current tools. Presently, the lack of baseline comparisons makes it difficult to put the claimed advantages into perspective.

**Questions:**

Your proposal could be used to split datasets based on their structural characteristics. Existing methods, such as scaffold splitting, are extensively used in drug discovery for similar purposes. How does your proposal align with, or differ from, these established techniques?

---

### Note · Authors · 2024-11-13

**Comment:**

Thank you every one for you constructive feedback.

**Withdrawal Confirmation:**

I have read and agree with the venue's withdrawal policy on behalf of myself and my co-authors.